# LEARNING WITH REFLECTIVE LIKELIHOODS

## ABSTRACT

Models parameterized by deep neural networks have achieved state-of-the-art results in many domains. These models are usually trained using the maximum likelihood principle with a finite set of observations. However, training deep probabilistic models with maximum likelihood can lead to the issue we refer to as *input forgetting*. In deep generative latent-variable models, input forgetting corresponds to *posterior collapse*—a phenomenon in which the latent variables are driven independent from the observations. However input forgetting can happen even in the absence of latent variables. We attribute input forgetting in deep probabilistic models to the *finite sample dilemma* of maximum likelihood. We formalize this problem and propose a learning criterion—termed *reflective likelihood*—that explicitly prevents input forgetting. We empirically observe that the proposed criterion significantly outperforms the maximum likelihood objective when used in classification under a skewed class distribution. Furthermore, the reflective likelihood objective prevents posterior collapse when used to train stochastic auto-encoders with amortized inference. For example in a neural topic modeling experiment, the reflective likelihood objective leads to better quantitative and qualitative results than the variational auto-encoder and the importance-weighted auto-encoder.

## 1 INTRODUCTION

Learning deep probabilistic models with maximum likelihood has led to many successes in density estimation, variational inference, and text and image generation (Dinh et al., 2016; Kingma & Dhariwal, 2018; Kingma & Welling, 2013; Rezende et al., 2014; Oord et al., 2016). However, maximum likelihood learning of deep models often causes the problem of *input forgetting*—so called because it corresponds to ignoring the input. We refer to an input here as any variable being conditioned upon—for example a covariate in supervised learning or a latent variable in deep generative latent variable models. Input forgetting corresponds to *latent variable collapse* in the context of latent variable models and has been discussed in several works (Burda et al., 2015; Bowman et al., 2015; Hoffman & Johnson, 2016; Chen et al., 2016; Sønderby et al., 2016; Zhao et al., 2017; Alemi et al., 2018; Dieng et al., 2018a). It also occurs when learning with Restricted Boltzmann Machines (RBMS) where all the hidden units of the RBM collapse by learning to only capture the bias in the visible units (Cho et al., 2011). Input forgetting makes posterior inference in deep generative models very difficult. In this paper we study input forgetting when the input is a latent variable and when the input is a covariate.

### 1.1 CONTRIBUTIONS

In this paper, we propose a learning criterion that does not suffer from the input forgetting problem induced by maximum likelihood when fitting deep probabilistic models. This criterion naturally arises from what we call the finite sample dilemma of maximum likelihood.

**The finite sample dilemma of maximum likelihood.** Consider a supervised learning setting where we have two random variables $\mathbf{x}$ and $\mathbf{y}$. Our goal is to learn the relationship between these two random variables by fitting a conditional parametric model $p_\theta(\mathbf{y} \mid \mathbf{x})$. The learning problem corresponds to finding $\theta^*$ that minimizes the risk function,

$$\mathcal{R}(\theta) = \mathbb{E}_{\mathbf{x} \sim p^*(\mathbf{x})} \left[ \mathbb{E}_{\mathbf{y} \sim p^*(\mathbf{y} \mid \mathbf{x})} \left[ l(\mathbf{y}, f_\theta(\mathbf{x})) \right] \right], \tag{1}$$

where $p^*$ denotes the population distribution and $l(\cdot, \cdot)$ is a loss function that measures how different $f_\theta(\mathbf{x})$ is from the true $\mathbf{y}$. Maximum likelihood corresponds to a log loss,

$$l(\mathbf{y}, f_\theta(\mathbf{x})) = -\log p_\theta(\mathbf{y} \mid \mathbf{x}) \quad \text{and} \quad \mathcal{R}_{mle}(\theta) = -\mathbb{E}_{\mathbf{x} \sim p^*(\mathbf{x})} \left[ \mathbb{E}_{\mathbf{y} \sim p^*(\mathbf{y} \mid \mathbf{x})} \left[ \log p_\theta(\mathbf{y} \mid \mathbf{x}) \right] \right]. \quad (2)$$

In practice we approximate $\mathcal{R}_{mle}(\theta)$ using a finite dataset $\mathcal{D} = (\mathbf{x}_n, \mathbf{y}_n)_{n=1}^N$ that contains $N$ realizations of the random variables $\mathbf{x}$ and $\mathbf{y}$. The sample-based maximum likelihood objective is

$$\tilde{\mathcal{L}}(\theta; \mathcal{D}) = \frac{1}{N} \sum_{n=1}^N \log p_\theta(\mathbf{y}_n | \mathbf{x}_n). \quad (3)$$

Now consider the risk of maximum likelihood under the same conditional model $p_\theta(\mathbf{y} \mid \mathbf{x})$ when $\mathbf{x}$ and $\mathbf{y}$ are independent,

$$\mathcal{R}'_{mle}(\theta) = -\mathbb{E}_{\mathbf{x} \sim p^*(\mathbf{x})} \left[ \mathbb{E}_{\mathbf{y} \sim p^*(\mathbf{y})} \left[ \log p_\theta(\mathbf{y} \mid \mathbf{x}) \right] \right]. \quad (4)$$

The corresponding sample-based maximum likelihood objective is

$$\tilde{\mathcal{L}}'(\theta; \mathcal{D}) = \frac{1}{N} \sum_{n=1}^N \frac{1}{N} \sum_{n'=1}^N \log p_\theta(\mathbf{y}_n | \mathbf{x}'_n). \quad (5)$$

The objectives $\tilde{\mathcal{L}}(\theta; \mathcal{D})$ and $\tilde{\mathcal{L}}'(\theta; \mathcal{D})$ in Eq. 3 and Eq. 5 are related as follows

$$\tilde{\mathcal{L}}(\theta; \mathcal{D}) = N \cdot \tilde{\mathcal{L}}'(\theta; \mathcal{D}) - \frac{1}{N} \sum_{n=1}^N \sum_{n' \neq n}^N \log p_\theta(\mathbf{y}_n | \mathbf{x}'_n). \quad (6)$$

Although there is no relationship between the two population risks $\mathcal{R}_{mle}(\theta)$ and $\mathcal{R}'_{mle}(\theta)$, there is a relationship between their sample-based estimates.

When maximizing the likelihood under finite data—i.e. maximizing $\tilde{\mathcal{L}}(\theta; \mathcal{D})$—the term $\tilde{\mathcal{L}}'(\theta; \mathcal{D})$ is also being maximized. However maximizing $\tilde{\mathcal{L}}'(\theta; \mathcal{D})$ leads to input forgetting as the underlying data being used do not encode any relationship between $\mathbf{x}$ and $\mathbf{y}$. To see this consider maximum likelihood with the same data but where each label $\mathbf{y}$ is also paired with all possible inputs $\mathbf{x}$. This version of the data does not contain any relationship between $\mathbf{x}$ and $\mathbf{y}$. Maximum likelihood using this data where $\mathbf{x}$ and $\mathbf{y}$ are independent corresponds to maximizing $\tilde{\mathcal{L}}'(\theta; \mathcal{D})$.

**A new learning criterion for supervised learning.** In light of the relationship in Eq. 6, we propose to explicitly discourage the independence behavior induced by maximizing $\tilde{\mathcal{L}}'(\theta; \mathcal{D})$. We do so by maximizing the *reflective likelihood* (RLL). Maximizing the RLL simultaneously maximizes the likelihood of the outputs under their true associated inputs and minimizes the average log-probability of these outputs when paired with all other possible inputs. On a classification under imbalance with the MNIST dataset, the classifiers trained with RLL outperform those trained with the usual maximum likelihood objective in all the instances of class imbalance. The performance were similar when the class distribution is uniform.

**A new family of stochastic auto-encoders for unsupervised learning.**

We extend the proposed RLL to unsupervised learning with stochastic auto-encoders and amortized inference. We call stochastic auto-encoders trained with the RLL objective Reflective Auto-Encoders (RAEs). RAEs do not suffer from the issue of posterior collapse—a phenomenon in which the estimated approximate posterior distribution of the latent variables reduces to the prior. We evaluate RAEs on a neural topic modelling experiment and found they outperform models learned with the maximum likelihood objective both quantitatively and qualitatively.

## 1.2 RELATED WORK.

Our work is closely related to two lines of work: penalized maximum likelihood and posterior inference in deep generative models.

Traditional penalized maximum likelihood methods such as the Lasso and $L_2$-norm regularization directly operate on the parameters of a model (Tibshirani, 1996; Hinton, 1987; Louizos et al., 2017).

These regularizers penalize the magnitude of the parameters and correspond to specific prior distributions on the parameters from the Bayesian perspective. In contrast several data-dependent regularizers have been proposed. As early as 1995, Bishop (1995) proposed to add noise to the input when training a neural network with stochastic gradient descent for maximum likelihood learning and showed this corresponds to a form of Tikhonov regularization. Several works have extended this to noise injection in the hidden units of a neural network (Srivastava et al., 2014; Maaten et al., 2013; Gal & Ghahramani, 2016; Wager et al., 2013; Dieng et al., 2018b). Our work relates to those data-dependent regularizers that are explicit—in the sense that the corresponding objective function can be written as the sum of the initial objective function and an additional regularization term.

Our work also relates to the literature on posterior inference in deep generative models. Latent variable models are ubiquitous in machine learning. They often involve intractable integrals that are approximated with Markov chain Monte Carlo or variational inference. Our work relates to variational methods for deep generative models (Kingma & Welling, 2013; Rezende et al., 2014). The problem that often occurs in these settings is the latent variables are driven independent to the observations thus rendering posterior inference meaningless. Several works have discussed this issue (Burda et al., 2015; Bowman et al., 2015; Hoffman & Johnson, 2016; Chen et al., 2016; Sønderby et al., 2016; Zhao et al., 2017; Alemi et al., 2018; Dieng et al., 2018a). In this paper we identify a possible cause to this problem and adopt a regularization approach to fix it.

## 2 Supervised Learning with Reflective Likelihoods

We consider the same setting as Section 1.1. We have observations $(\mathbf{x}_n, \mathbf{y}_n)_{n=1}^N$ and our goal is to fit a parametric model $p_\theta(\mathbf{y} \mid \mathbf{x})$ to this data. As stated in Section 1.1, the usual maximum likelihood objective might lead to parameters $\theta$ that do not capture the true relationship between $\mathbf{x}$ and $\mathbf{y}$. In this section we propose the RLL criterion that encourages parameter settings under which $\mathbf{y}$ is more likely when conditioning on its corresponding $\mathbf{x}$ than under the average of all the conditional probabilities of $\mathbf{y}$ given all possible inputs. The RLL criterion has two forms.

**Weak reflective likelihood.** The weaker form of RLL minimizes $\tilde{\mathcal{L}}'(\theta; \mathcal{D})$ while maximizing the usual sample-based approximation $\tilde{\mathcal{L}}(\theta; \mathcal{D})$,

$$\tilde{\mathcal{L}}_{RLL}^{\text{weak}}(\theta; \mathcal{D}) = \frac{1}{N} \sum_{n=1}^N \left[ \log p_\theta(\mathbf{y}_n \mid \mathbf{x}_n) - \frac{\alpha_n}{N} \sum_{n'=1}^N \log p_\theta(\mathbf{y}_n \mid \mathbf{x}_{n'}) \right], \quad (7)$$

where $\alpha_n \geq 0$ is a hyperparameter trading off these two terms. We will discuss how to set $\alpha_n$ at the end of this section. The objective $\tilde{\mathcal{L}}_{RLL}^{\text{weak}}(\theta; \mathcal{D})$ provides us with a way to explicitly prefer $\mathcal{R}_{mle}(\theta)$ in Eq. (1) over $\mathcal{R}'_{mle}(\theta)$ in Eq. (4), over an entire training set. The objective $\tilde{\mathcal{L}}_{RLL}^{\text{weak}}(\theta; \mathcal{D})$, however, may capture the dependence between the input and the output for some examples but not necessarily for all examples. This is because the second term in Eq. 7 can be driven to its minimal value by setting the term inside the log to zero for one example. We tackle this issue by proposing a stronger version of RLL.

**Strong reflective likelihood.** The stronger version of RLL favors parameters that capture the relationship between $\mathbf{x}$ and $\mathbf{y}$ by discouraging the independence behavior induced by maximization of $\tilde{\mathcal{L}}'(\theta; \mathcal{D})$,

$$\tilde{\mathcal{L}}_{RLL}^{\text{strong}}(\theta; \mathcal{D}) = \frac{1}{N} \sum_{n=1}^N \left[ \log p_\theta(\mathbf{y}_n \mid \mathbf{x}_n) - \alpha_n \log \frac{1}{N} \sum_{n'=1}^N p_\theta(\mathbf{y}_n \mid \mathbf{x}_{n'}) \right]. \quad (8)$$

The second term comes from an application of Jensen's inequality,

$$\mathcal{R}'_{mle}(\theta) \leq \mathbb{E}_{\mathbf{y} \sim p^*(\mathbf{y})} \left[ \log \mathbb{E}_{\mathbf{x} \sim p^*(\mathbf{x})} \left[ p^\theta(\mathbf{y} \mid \mathbf{x}) \right] \right],$$

and using the dataset $\mathcal{D}$ to approximate this upper bound. The inner term of this upper bound is what we call a *reflective probability*,

$$p_\theta^{\text{refl}}(y) = \mathbb{E}_{\mathbf{x} \sim p^*(\mathbf{x})} p_\theta(\mathbf{y} \mid \mathbf{x}). \quad (9)$$

This reflective probability can be used as a diagnostic measure of whether a trained model has captured the dependence between the input and output for *each* pair. The sample-based approximation of $p_\theta^{\text{refl}}(y)$ is what appears inside the logarithm in the second term in Eq. 8.

Unlike $\tilde{\mathcal{L}}_{RLL}^{\text{weak}}$, this stronger objective function $\tilde{\mathcal{L}}_{RLL}^{\text{strong}}$ explicitly decreases the reflective probability of each individual example $(\mathbf{x}, \mathbf{y})$. From here on out we focus on $\tilde{\mathcal{L}}_{RLL}^{\text{strong}}$ and refer to it as RLL.

**Stochastic Approximation of RLL.** The objective in Eq. 8 is not amenable to large datasets. First, when the sample size $N$ is large computing $\tilde{\mathcal{L}}_{RLL}^{\text{strong}}(\theta; \mathcal{D})$ is intractable. Second even if we use a small subset of data we are still left with approximating $p_\theta^{\text{refl}}(y)$ which is also intractable for large $N$. To enable maximizing RLL with large datasets, we propose to maximize

$$\tilde{\mathcal{L}}_{RLL}^{\text{strong}}(\theta; \mathcal{M}, \mathcal{M}') = \frac{1}{|\mathcal{M}|} \sum_{(\mathbf{x}, \mathbf{y}) \in \mathcal{M}} \left[ \log p_\theta(\mathbf{y} \mid \mathbf{x}) - \alpha_n \log \frac{1}{|\mathcal{M}'|} \sum_{(\mathbf{x}', \mathbf{y}') \in \mathcal{M}'} p_\theta(\mathbf{y} \mid \mathbf{x}') \right], \quad (10)$$

where $\mathcal{M}$ and $\mathcal{M}'$ are two minibatch of data samples. In practice we improve the efficiency by constructing the second minibatch $\mathcal{M}'$ using the first minibatch $\mathcal{M}$. For each pair $(\mathbf{x}, \mathbf{y}) \in \mathcal{M}$, we uniformly select $K$ examples from $\mathcal{M}$ at random to construct $\mathcal{M}'$.

**Choice of penalty $\alpha_n$.** We investigate two strategies to set the cofficient $\alpha_n$. The first strategy simply sets $\alpha_n$ to a global fixed coefficient $\alpha_n = \alpha_0$, where $\alpha_0$ is a hyperparameter. The second strategy on the other hand dynamically adapts $\alpha_n$ for each training example $(\mathbf{x}_n, \mathbf{y}_n)$ according to:

$$\alpha_n = \alpha(\mathbf{x}_n, \mathbf{y}_n, \theta) = \begin{cases} \alpha_0 & \text{if } \log p_\theta(\mathbf{y}_n \mid \mathbf{x}_n) \leq \log \hat{p}_\theta^{\text{refl}}(\mathbf{y}_n) \\ 0 & \text{otherwise} \end{cases} \quad (11)$$

The second strategy is motivated by the intuition that the probability assigned to a correctly paired example $(\mathbf{x}_n, \mathbf{y}_n)$ by a model $p_\theta$ should be higher than the average of the probabilities assigned to incorrectly paired examples $(\mathbf{x}, \mathbf{y}_n)$ for all $\mathbf{x} \sim p^*(\mathbf{x})$. It is however unclear how much higher the probability of a correctly paired example $p_\theta(\mathbf{y}_n \mid \mathbf{x}_n)$ should be over the reflective probability $p_\theta^{\text{refl}}(\mathbf{y}_n)$, and we do not want to increase the difference if the former is already greater than the latter.

# 3 Unsupervised Learning with Reflective Likelihoods

We extend RLL to unsupervised learning. We do so using the auto-encoding framework (Hinton & Salakhutdinov, 2006; Vincent et al., 2008; Kingma & Ba, 2014; Rezende et al., 2014). We first discuss stochastic auto-encoders and posterior collapse before proposing RAEs—a family of stochastic auto-encoders that do not suffer from the posterior collapse issue.

**Stochastic auto-encoders and posterior collapse.** We position ourselves in the setting where there are global parameters $\theta$ and one latent variable $\mathbf{z}$ for every observation $\mathbf{y}$. An observation $\mathbf{y}$ is generated by first drawing a latent variable $\mathbf{z}$ from some prior distribution $p(\mathbf{z})$—that we assume fixed—and then sampling $\mathbf{y}$ from the conditional distribution of $\mathbf{y}$ given $\mathbf{z}$. This conditional distribution $p_\theta(\mathbf{y} \mid \mathbf{z})$ is parameterized by $\theta$. We are concerned with learning the parameters $\theta$ in the presence of the latent variables. Maximum likelihood corresponds to maximizing

$$\mathcal{L}(\theta; p^*) = \mathbb{E}_{\mathbf{y} \sim p^*(\mathbf{y})} \log p_\theta(\mathbf{y}) = \mathbb{E}_{\mathbf{y} \sim p^*(\mathbf{y})} \left[ \log \int_z p_\theta(\mathbf{y} \mid \mathbf{z}) p(\mathbf{z}) \mathrm{d}\mathbf{z} \right]. \quad (12)$$

The likelihood $p_\theta(\mathbf{y} \mid \mathbf{z})$ is represented as a powerful deep neural network that takes $\mathbf{z}$ as input. This objective Eq. 12 is intractable. Stochastic auto-encoders posit a distribution $q_\phi(\mathbf{z} \mid \mathbf{y})$ over the latents and either maximize (Burda et al., 2015)

$$\mathcal{L}(\theta, \phi; p^*) = \mathbb{E}_{\mathbf{y} \sim p^*(\mathbf{y})} \left[ \log \mathbb{E}_{\mathbf{z} \sim q_\phi(\mathbf{z} \mid \mathbf{y})} \left[ \frac{p_\theta(\mathbf{y} \mid \mathbf{z}) p(\mathbf{z})}{q_\phi(\mathbf{z} \mid \mathbf{y})} \right] \right], \quad (13)$$

or its lower bound (Kingma & Ba, 2014; Rezende et al., 2014)

$$\underline{\mathcal{L}}(\theta, \phi; p^*) = \mathbb{E}_{\mathbf{y} \sim p^*(\mathbf{y})} \left[ \mathbb{E}_{\mathbf{z} \sim q_\phi(\mathbf{z} \mid \mathbf{y})} \log \left[ \frac{p_\theta(\mathbf{y} \mid \mathbf{z}) p(\mathbf{z})}{q_\phi(\mathbf{z} \mid \mathbf{y})} \right] \right]. \tag{14}$$

The distribution $q_\phi(\mathbf{z} \mid \mathbf{y})$ is parameterized by a deep neural network that takes the observation $\mathbf{y}$ as input.

As discussed in several works, the objective in Eq. 14 is prone to posterior collapse. It can be rewritten as

$$\mathcal{L}^{\mathrm{vae}}(\theta, \phi; p^*) = \mathbb{E}_{\mathbf{y} \sim p^*(\mathbf{y})} \left[ \mathbb{E}_{\mathbf{z} \sim q_\phi(\mathbf{z} \mid \mathbf{y})} \log p_\theta(\mathbf{y} \mid \mathbf{z}) - \mathrm{KL}(q_\phi(\mathbf{z} \mid \mathbf{y}) \| p(\mathbf{z})) \right], \tag{15}$$

What effectively happens is that the KL term quickly collapses to zero early on during optimization, leading to an approximate posterior that does not capture the data.

The importance-weighted auto-encoder (IWAE, Burda et al., 2015) was introduced to learn more expressive approximate posteriors by directly optimizing the objective in Eq. (13) using $K$ samples from the approximate posterior,

$$\mathcal{L}^{\mathrm{iwae}}(\theta; p^*) = \mathbb{E}_{\mathbf{y} \sim p^*(\mathbf{y})} \left[ \log \frac{1}{K} \sum_{k=1}^{K} \left[ \frac{p_\theta(\mathbf{y} \mid \mathbf{z}_k) p(\mathbf{z}_k)}{q_\phi(\mathbf{z}_k \mid \mathbf{y})} \right] \right]. \tag{16}$$

Burda et al. (2015) show this leads to less posterior collapse. However, the IWAE can still lead to posterior collapse due to the finite sample dilemma of maximum likelihood discussed in Section 1.

**Reflective auto-encoders.** We propose a new class of stochastic auto-encoders that prevent posterior collapse by maximizing the RLL. We reconsider the original population risk functions for maximum likelihood under both paired data,

$$\mathcal{R}_{mle}(\theta) = -\mathbb{E}_{\mathbf{x} \sim p^*(\mathbf{x})} \left[ \mathbb{E}_{\mathbf{y} \sim p^*(\mathbf{y} \mid \mathbf{x})} \left[ \log p_\theta(\mathbf{y} \mid \mathbf{x}) \right] \right]$$

and its upper bound under independent data,

$$\mathcal{R}'_{mle}(\theta) \leq \mathbb{E}_{\mathbf{y} \sim p^*(\mathbf{y})} \left[ \log \mathbb{E}_{\mathbf{x} \sim p^*(\mathbf{x})} \left[ p_\theta(\mathbf{y} \mid \mathbf{x}) \right] \right].$$

The RLL in Eq. 8 is a sample-based approximation of

$$\mathcal{L}^{sup}_{RLL}(\theta; p^*) = \mathbb{E}_{\mathbf{x} \sim p^*(\mathbf{x})} \mathbb{E}_{\mathbf{y} \sim p^*(\mathbf{y} \mid \mathbf{x})} \left[ \log p_\theta(\mathbf{y} \mid \mathbf{x}) - \alpha \log \mathbb{E}_{\mathbf{x}' \sim p^*(\mathbf{x}')} \left[ p_\theta(\mathbf{y} \mid \mathbf{x}') \right] \right].$$

Note this is simply the difference between the original maximum likelihood objective and the log reflective probability. In the case of unsupervised learning this corresponds to

$$\mathcal{L}^{unsup}_{RLL}(\theta) = \mathbb{E}_{\mathbf{y} \sim p^*(\mathbf{y})} \left[ \log p_\theta(\mathbf{y}) - \alpha \log \mathbb{E}_{\mathbf{y}' \sim p^*(\mathbf{y})} \left[ p_\theta(\mathbf{y} \mid \mathbf{y}') \right] \right], \tag{17}$$

where we define $p_\theta(\mathbf{y} \mid \mathbf{y}')$ as

$$p_\theta(\mathbf{y} \mid \mathbf{y}') = \int p_\theta(\mathbf{y} \mid \mathbf{z}) p_\theta(\mathbf{z} \mid \mathbf{y}') d\mathbf{z} = \mathbb{E}_{p_\theta(\mathbf{z} \mid \mathbf{y}')}[p_\theta(\mathbf{y} \mid \mathbf{z})].$$

Using this definition we rewrite the objective as

$$\mathcal{L}^{unsup}_{RLL}(\theta) = \mathbb{E}_{\mathbf{y} \sim p^*(\mathbf{y})} \left[ \log \mathbb{E}_{\mathbf{z} \sim p(\mathbf{z})} p_\theta(\mathbf{y} \mid \mathbf{z}) - \alpha \log \mathbb{E}_{\mathbf{y}' \sim p^*(\mathbf{y})} \mathbb{E}_{p_\theta(\mathbf{z} \mid \mathbf{y}')}[p_\theta(\mathbf{y} \mid \mathbf{z})] \right]. \tag{18}$$

The expectations in Eq. 18 are intractable. In this paper, we propose to approximate them using importance weighting (Burda et al., 2015). For that we define a parametric proposal distribution $q_\phi(\mathbf{z} \mid \mathbf{y})$—we make the conditioning on $\mathbf{y}$ explicit to account for recognition networks as proposal distributions. We now write the different expectations as

$$\mathbb{E}_{\mathbf{z} \sim p(\mathbf{z})}\left[ p_\theta(\mathbf{y} \mid \mathbf{z}) \right] \approx \hat{p}_\theta(\mathbf{y}) = \frac{1}{K} \sum_{k=1}^{K} \omega(\mathbf{y}, \mathbf{z}^k) p_\theta(\mathbf{y} \mid \mathbf{z}^k) \quad \text{where} \quad \mathbf{z}^k \sim q_\phi(\mathbf{z} \mid \mathbf{y});$$

$$\text{and} \quad \mathbb{E}_{\mathbf{y}' \sim p^*(\mathbf{y})} \mathbb{E}_{\mathbf{z} \sim p_\theta(\mathbf{z} \mid \mathbf{y}')} \left[ p_\theta(\mathbf{y} \mid \mathbf{z}) \right] \approx \mathbb{E}_{\mathbf{y}' \sim p^*(\mathbf{y})} \left( \sum_{k=1}^{K} \upsilon(\mathbf{y}', \mathbf{z}^k) p_\theta(\mathbf{y} \mid \mathbf{z}^k) \right)$$

where $\mathbf{z}^k \sim q_\phi(\mathbf{z} \,|\, \mathbf{y}')$. The expensive expectation over $p^*(\mathbf{y}')$ is dealt with as before by using a small random subset of training examples. The importance weights $\omega(\mathbf{y}, \mathbf{z}^k)$ and $v(\mathbf{y}', \mathbf{z}^k)$ are computed as

$$\omega(\mathbf{y}, \mathbf{z}^k) = \exp(\tilde{\omega}(\mathbf{y}, \mathbf{z}^k)) \quad \text{and} \quad \tilde{\omega}(\mathbf{y}, \mathbf{z}^k) = \log p(\mathbf{z}^k) - \log q_\phi(\mathbf{z}^k \,|\, \mathbf{y}).$$

$$v(\mathbf{y}', \mathbf{z}^k) = \frac{\exp(\tilde{v}(\mathbf{y}', \mathbf{z}^k))}{\sum_{s=1}^K \exp(\tilde{v}(\mathbf{y}', \mathbf{z}^s))} \quad \text{and} \quad \tilde{v}(\mathbf{y}', \mathbf{z}^k) = \log p_\theta(\mathbf{y}' \,|\, \mathbf{z}^k) + \log p(\mathbf{z}^k) - \log q_\phi(\mathbf{z}^k \,|\, \mathbf{y}').$$

Maximizing the objective in Eq. 18 encourages the proposal $q_\phi$ to output a distinct approximate posterior distribution for each input $\mathbf{y}$. We conjecture this helps avoid posterior collapse. We verify this later in our empirical study.

## 4    CONNECTIONS TO RANKING

Collobert et al. (2011) proposed a pair-wise ranking loss for natural language processing. This ranking loss is defined as

$$\mathcal{L}_{\text{ranking}} = - \left[ m - f^\theta(s^{\text{pos}}) + f^\theta(s^{\text{neg}}) \right]_+ , \tag{19}$$

where $f^\theta$ is a parametric scoring function, and $s^{\text{pos}}$ and $s^{\text{neg}}$ are positive and negative examples respectively. $s^{\text{pos}}$ is often simply a training example such as a sentence, whereas $s^{\text{neg}}$ is a sample from a distribution conditioned on $s^{\text{pos}}$. The hyperparameter $m$ is the margin.

Consider the RLL objective in Eq. (8). Consider the choice of $\alpha_n$ in Eq. (11). Using stochastic approximation with a single sample $(\mathbf{x}, \mathbf{y})$ for the likelihood term and a single sample $(x', y)$ for the reflective probability we can rewrite the objective $\tilde{\mathcal{L}}_{RLL}^{\text{strong}}(\theta; \mathcal{D})$ as

$$\begin{aligned} \tilde{\mathcal{L}}_{RLL}^{\text{strong}}(\theta; \mathcal{D}) &= \log p_\theta(\mathbf{y} \,|\, \mathbf{x}) - \alpha_0 \log p_\theta(\mathbf{y} \,|\, \mathbf{x}') \\ &= (1 - \alpha_0) \log p^\theta(\mathbf{y} \,|\, \mathbf{x}) - \alpha_0 \left[ 0 - \log p_\theta(\mathbf{y} \,|\, \mathbf{x}) + \log p_\theta(\mathbf{y} \,|\, \mathbf{x}') \right]_+ . \end{aligned}$$

This reveals that RLL is a convex sum between the usual maximum likelihood objective and the ranking loss in Eq. (19), modulated by the global coefficient $\alpha_0$. It directly tackles the finite sample dilemma discussed in Section 1 by maximizing the likelihood while ensuring that the correctly paired examples are better scored than incorrectly paired examples. This in turn maximizes the dependence between the inputs and outputs.

## 5    APPLICATIONS

In this section we apply our proposed method to two different problems: MNIST digit classification under imbalance and neural topic modeling. In classification under imbalance it is hard to learn features of rare classes. Because RLL promotes a stronger dependence between inputs and outputs we expect it to perform better than the maximum likelihood objective in the presence of imbalance. We also consider neural topic models that use stochastic auto-encoders. Posterior collapse is particularly severe in text modeling. Neural topic models suffer from this problem (Srivastava & Sutton, 2017). One manifestation of posterior collapse in neural topic modeling is that all the dimensions of the topic matrix collapse to the same topic which in turn contains words that are not necessarily related to each other (Miao et al., 2016; Srivastava & Sutton, 2017). On both applications we found RLL yields better performance both quantitatively and qualitatively. It learns more useful features for rare classes in classification under imbalance on MNIST as evidenced by higher F1 scores (See fig. 1.) Finally it learns more meaningful latent variables as evidenced by lower perplexity on document completion (See Table 2.) We also found that the choice of $\alpha_n$ is dependent on the application. For classification we found a fixed schedule for alpha to perform best. For neural topic modeling we found an adaptive schedule to perform best. The reported results for RLL correspond to the best schedule for $\alpha_n$.

**Figure 1:** Histogram of classification F1 scores for MLE and RLL. **Left:** Uniform distribution D1. **Right:** Imbalanced distribution D10. Performance of MLE and RLL on D1 is similar. However RLL outperforms MLE by a significant margin for the imbalanced distribution. This gain in performance comes from how well RLL performs on rare classes. For digits 2, 6, and 8 both MLE and RLL have 0 F1 scores.

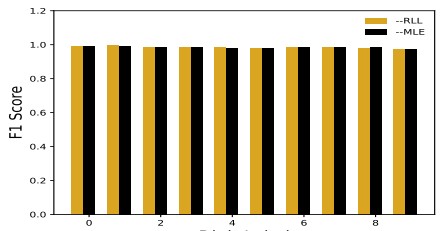 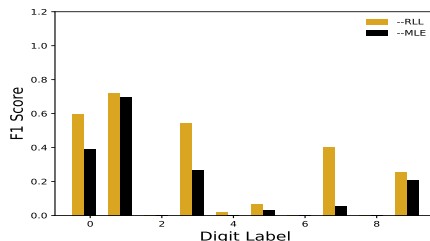

**Table 1:** F1 scores and accuracies on (a) the original test set and (b) matched test sets. Across the board, the classifiers trained with the proposed reflective log-likelihood perform similarly to or outperform those trained with maximum log-likelihood.

(a) Original Test Set

| Method | Metric | D1 | D2 | D3 | D4 | D5 | D6 | D7 | D8 | D9 | D10 |
|--------|--------|------|------|------|------|------|------|------|------|------|------|
| MLE | Acc | 98.5 | 89.0 | 77.7 | 68.5 | 65.4 | 55.9 | 45.5 | 31.9 | 28.0 | 21.1 |
| RLL | Acc | **98.6** | **92.0** | **83.3** | **70.2** | **71.7** | **59.1** | **48.8** | **35.6** | **33.8** | **31.1** |
| MLE | F1 | 98.5 | 84.5 | 69.3 | 57.9 | 57.8 | 43.9 | 32.2 | 19.5 | 20.6 | 17.5 |
| RLL | F1 | **98.6** | **92.6** | **80.9** | **60.7** | **65.9** | **48.4** | **37.2** | **22.8** | **27.4** | **27.0** |

(b) Matched Test Set

| Method | Metric | D1 | D2 | D3 | D4 | D5 | D6 | D7 | D8 | D9 | D10 |
|--------|--------|------|------|------|------|------|------|------|------|------|------|
| MLE | Acc | **99.2** | 90.9 | 80.8 | 72.4 | 66.1 | 55.7 | 45.1 | 32.6 | 29.2 | 19.5 |
| RLL | Acc | 99.0 | **93.9** | **85.1** | **74.5** | **74.1** | **57.3** | **48.1** | **37.5** | **34.1** | **28.8** |
| MLE | F1 | **99.2** | 87.0 | 73.1 | 62.7 | 57.5 | 43.7 | 31.2 | 20.7 | 22.8 | 15.9 |
| RLL | F1 | 99.0 | **94.3** | **83.3** | **66.2** | **68.8** | 45.9 | **36.1** | **24.1** | **27.8** | **24.8** |

## 5.1 CLASSIFICATION UNDER CLASS IMBALANCE

We start with MNIST which has 5,000 training examples per class, totalling 50,000 training examples. We refer to this original dataset as **D**0. We build 9 additional training sets. Each additional training set **D**$k$ leaves only 5 training examples for each class $j < k$. See Table 3 in the Appendix for all the class distributions. The validation set in each case is constructed by collecting all the data points that are left from the training set. We train a classifier on each of these constructed training sets and test it on both the **original test set**, in which all the classes are uniformly represented, and the **matched test set**, in which the class distribution matches that of the corresponding training set. We measure both F1 score and accuracy for each setting.

As a classifier, we use a deep convolutional network (LeCun et al., 1995). It has two convolutional layers and two fully connected layers. Each convolutional layer consists of $5 \times 5$ convolution with stride 1 followed by $2 \times 2$ max pooling and a rectified linear unit (ReLU, Glorot et al., 2011). The output from the final convolutional layer is flattened and processed by two fully-connected layers of 50 units. We train the classifier using the Adam optimizer with dropout regularization. We find the number of samples to estimate the second term in RLL by maximizing accuracy on the validation set. We found that in most cases one sample suffices. We choose $\alpha_0$ to maximize accuracy on the validation set. We found $\alpha_0 = 0.5$ to be the best.

In Table 1, we see that the classifiers trained with RLL always outperform those trained with the usual maximum likelihood criterion when the class distribution is imbalanced. This is the case both in terms of F1 score and accuracy on both the original (balanced) test set as well as the matched test set.

**Table 2:** Perplexity and KL divergence computed from the neural topic model trained on the 20NewsGroup dataset. $K$ denotes the number of posterior samples used during training. ($\star$) The difference between VAE ($K = 1$) and IWAE ($K = 1$) comes from the fact that we use an analytical solution to the KL divergence in the case of VAE and sample-based approximation in the case of IWAE.

| Criterion | Autoencoder | $K$ | Full PPL↓ | Completion PPL↓ | KL($q_\phi(\mathbf{z} \,|\, \mathbf{y}) \,\|\, p(\mathbf{z})$)↑ |
|-----------|-------------|-----|-----------|-----------------|------------------------------------------------------------------|
| MLE | VAE | 1 | 841 | 911 | 2.3 |
| MLE | IWAE | 1 | 820 | 884 | 2.8 |
| RLL | RAE | 1 | **809** | **875** | **3.0** |
| MLE | VAE | 100 | 818 | 886 | 3.1 |
| MLE | IWAE | 100 | 780 | 838 | 3.5 |
| RLL | RAE | 100 | **763** | **822** | **4.0** |

## 5.2 Neural Topic Modelling

We use the 20NewsGroup (Joachims, 1996) for topic modelling. 20NewGroup consists of 11,314 training and 7,531 test articles. The vocabulary size is 2,000. We follow the standard preprocessing steps which include tokenization and the removal of non-UTF-8 characters and English stop words (Miao et al., 2016; Srivastava & Sutton, 2017; Card et al., 2017).

We train a neural topic model, which is an extension of latent Dirichlet allocation (LDA, Blei et al., 2003) with amortized inference. This is effectively a stochastic autoencoder with a bag-of-word input and output. The inference network $q_\phi$ of this auto-encoder is a two-layer feed-forward neural network with 100 $\tanh$ units each. The output of this network is projected to a 15-dimensional vector normalized by softmax to compute the topic proportion. The likelihood $p_\theta$ is a three-layer feed-forward neural network that maps this topic proportion to a bag-of-words. Our architecture choice is based on the earlier observation by Srivastava & Sutton (2017) that this model is particularly prone to posterior collapse. We test both the VAE (see Eq. (15)) and the IWAE (see Eq. (16)) as baselines. These are two auto-encoders trained with different forms of the maximum likelihood objective. We use Adam (Kingma & Ba, 2014) and train each model for 200 epochs with a fixed learning rate of 0.002. For RLL, we use 5 samples to estimate the reflective probability. We choose an adaptive schedule for $\alpha_n$.

We measure both perplexity and the KL divergence between the estimated posterior distribution $q_\phi$ and the prior distribution $p(\mathbf{z}) = \mathcal{N}(\mathbf{z}; 0, 1^2)$. The KL has been used as a metric for measuring posterior collapse. For perplexity, we report both the perplexity on the test set (Full PPL) and the document completion perplexity (Completion PPL, Wallach et al., 2009). Both types of perplexities are computed using importance sampling. Document completion consists in holding out some words for each document in the test set to compute the topic proportions and then evaluating perplexity on the remaining words of each document using the topic proportions learned with the held out words. It is a good way to assess the quality of the learned latent variables in topic modeling. We held out the first half of each document to compute the topic proportions and evaluated perplexity on the second half.

Similarly to the classification experiment, we observe that RAE—the stochastic auto-encoder trained with RLL—always outperforms the neural topic models trained with a maximum likelihood objective (VAE and IWAE) (see Table 2.) In addition to having a lower perplexity, RLL yields slightly higher KL divergence scores—which suggests that RLL indeed improves upon VAE and IWAE in terms of posterior collapse. We verify this further by looking at the learned topics; RLL the topics learned by RLL are as good as the topics learned by Latent Dirichlet Allocation (LDA). This is not the case for the VAE. IWAE also yields good topics but they are less sharp than the ones learned by RAE. These topics can be found in the appendix.

## 6 CONCLUSION

In this paper, we identified the finite sample dilemma when fitting probabilistic models with maximum likelihood using a finite set of observations. We conjectured this to be the cause of the input forgetting issue often encountered when fitting deep probabilistic models using maximum likelihood. We then proposed RLL—a new learning criterion that mitigates the input forgetting issue. RLL encourages a strong dependence between observations and their conditioning variables. This in turn results in better performance both in classification under imbalance and neural topic modeling with stochastic auto-encoders.

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

## 7 APPENDIX

**Table 3:** Class distributions using the MNIST dataset. There are 10 class—one class for each of the 10 digits in MNIST. The distribution D1 is uniform and the other distributions correspond to different imbalance settings as given by the proportions in the table. Note these proportions might not sum to one exactly because of rounding.

| Dist | 0 | 1 | 2 | 3 | 4 | 5 | 6 | 7 | 8 | 9 |
|------|------|------|------|------|------|------|------|------|------|------|
| D1 | 0.1 | 0.1 | 0.1 | 0.1 | 0.1 | 0.1 | 0.1 | 0.1 | 0.1 | 0.1 |
| D2 | $1e^{-3}$ | 0.11 | 0.11 | 0.11 | 0.11 | 0.11 | 0.11 | 0.11 | 0.11 | 0.11 |
| D3 | $1e^{-3}$ | $1e^{-3}$ | 0.12 | 0.12 | 0.12 | 0.12 | 0.12 | 0.12 | 0.12 | 0.12 |
| D4 | $1e^{-3}$ | $1e^{-3}$ | $1e^{-3}$ | 0.14 | 0.14 | 0.14 | 0.14 | 0.14 | 0.14 | 0.14 |
| D5 | $1e^{-3}$ | $1e^{-3}$ | $1e^{-3}$ | $1e^{-3}$ | 0.17 | 0.17 | 0.17 | 0.17 | 0.17 | 0.17 |
| D6 | $1e^{-3}$ | $1e^{-3}$ | $1e^{-3}$ | $1e^{-3}$ | $1e^{-3}$ | 0.20 | 0.20 | 0.20 | 0.20 | 0.20 |
| D7 | $1e^{-3}$ | $1e^{-3}$ | $1e^{-3}$ | $1e^{-3}$ | $1e^{-3}$ | $1e^{-3}$ | 0.25 | 0.25 | 0.25 | 0.25 |
| D8 | $1e^{-3}$ | $1e^{-3}$ | $1e^{-3}$ | $1e^{-3}$ | $1e^{-3}$ | $1e^{-3}$ | $1e^{-3}$ | 0.33 | 0.33 | 0.33 |
| D9 | $1e^{-3}$ | $1e^{-3}$ | $1e^{-3}$ | $1e^{-3}$ | $1e^{-3}$ | $1e^{-3}$ | $1e^{-3}$ | $1e^{-3}$ | 0.49 | 0.49 |
| D10 | $1e^{-3}$ | $1e^{-3}$ | $1e^{-3}$ | $1e^{-3}$ | $1e^{-3}$ | $1e^{-3}$ | $1e^{-3}$ | $1e^{-3}$ | $1e^{-3}$ | 0.99 |

**Table 4:** Top ten words of five randomly selected topics for different models on the 20NewsGroup dataset. Overall RAE learns topics as good as LDA—a non-neural network based model that does not suffer from posterior collapse. The VAE suffers from collapse and learns topics that are meaningless. The IWAE learns better topics than the VAE but has two topics (topics 3 and 5) that are worse than RAE.

| Setting | Topics |
|---------|--------|
| LDA | israel israeli jews arab state peace land jewish write policy
pay tax insurance money write care year article health rate
car drive engine buy write speed article light dealer driver
offer sale condition mouse best old excellent tape month trade
game team year win play season player fan write baseball |
| VAE | write article thanks want try need help buy work really
write article thanks thing really look buy drive gun problem
thanks run work write problem program software drive buy computer
write article buy drive thanks problem car try want help
armenians armenian kill turkish say child war government live attack |
| IWAE | game play team player year write article better win baseball
god christian jesus faith bible truth church christ believe christianity
write article car thanks buy problem game bike team player
armenians armenian turks turkish government israel state genocide attack serve
thanks buy write car article phone appreciate drive sale run |
| RAE | god christian jesus life faith church believe bible christ christianity
game team play player win score year season hit toronto
gun weapon say kill police come carry crime health criminal
card drive windows mode driver problem pc disk printer scsi
government key clipper chip secure encryption escrow law enforcement security |

