# OpenReview forum: "Learning with Reflective Likelihoods"
_ICLR.cc/2019/Conference_

### Official Review · AnonReviewer3 · 2018-10-30
**potentially interesting idea, but very confusing in current form**

**Rating:** 3
**Confidence:** 4

**Review:**

The paper proposes a modification of maximum likelihood estimation that encourages estimated predictive/conditional models p(z|x) to have low entropy and/or to maximize mutual information between z and x under the model p_{data}(x)p_{model}(z|x).

There is pre-existing literature on encouraging low entropy / high mutual information in predictive models, suggesting this can indeed be a good idea. The experiments in the current paper are preliminary but encouraging. However, the setting in which the paper presents this approach (section 2.1) does not make any sense. Also see my previous comments.

- Please reconsider your motivation for the proposed method. Why does it work? Also please try to make the connection with the existing literature on minimum entropy priors and maximizing mutual information.

- Please try to provide some guarantees for the method. Maximum likelihood estimation is consistent: given enough data and a powerful model it will eventually do the right thing. What will your estimator converge to for infinite data and infinitely powerful models?

---

> ### Author Response · Authors · 2018-11-27
> **Response to Reviewer 3**
>
> Thank you for reviewing our paper. We are very glad you find the idea potentially interesting and that the experimental results are encouraging.
>
> In the first version of our paper, we made a wrong statement when justifying our proposed objective. This made the paper very confusing as you mentioned in your review. In light of this feedback, we refactored the paper with a new perspective on why regularizing maximum likelihood with the log reflective probability is a good thing to do. The results did not change because the method is the same. It is the explanation behind why our proposed method works that changed. We replaced the section 2 of version 1 with the paragraph titled “the finite sample dilemma of maximum likelihood” in the introduction. We chose to add this paragraph in the introduction for sake of clarity and because we want the reader to grasp the intuition behind the proposed objective early on.
>
> We hope these changes address your earlier concerns.
>
> Regarding your question on connections to mutual information: our objective increases the dependence between inputs and outputs as evidenced in the empirical study section and as motivated by the finite sample dilemma of maximum likelihood. In this sense the RLL objective is implicitly related to mutual information which is a measure of dependence. However, there is no mathematical equation that directly relates RLL and mutual information.
>
> Regarding your question on asymptotics: for infinite data the criterion in Eq. 8 (which is the RLL...the one we use) converges in probability to the difference between the true maximum likelihood objective (the one using expectations under the population distribution) and the log reflective-probability (we define this in the paper...it is some marginal over the output y). In this sense our finite-data objective in Eq. 8 is a consistent estimator of the true objective. You get this result using the law of large numbers and continuity of logarithm.
>
> We hope we have addressed your concerns. Please let us know if you have further remarks.

---

> > ### Author Response · Authors · 2018-12-03
> > **Have we addressed your concerns?**
> >
> > Dear Reviewer 3,
> >
> > Thanks again for your review. We were wondering if we have addressed all your concerns and if you have further comments.

---

### Official Review · AnonReviewer2 · 2018-10-31
**the paper is technically flawed**

**Rating:** 2
**Confidence:** 4

**Review:**

This paper is technically flawed. Here are three key equations from Section 2. The notations are simplified for textual presentation:  d – p_data; d(y|x) – p_d(y|x); m(y|x) – p_theta(y|x)

max E_x~d E_y~d(y|x) [ log m (y|x) ]                 				               (1)
max E_x~d { E_y~d(y|x) ) [ log d(y|x) ]}  -  E_y~d(y|x) [ log m (y|x) ]}        (2)
max { E_y~d [  log  (y) ]  -  E_y~d  log E_x~d(x|y) [ m (y|x) ]}                        (3)

First error is that the “max” in (2) and (3) should be “min”. I will assume this minor error is corrected in the following.
The equivalence between (1) and (2) is correct and well-known. The reason is that the first entropy term  in (2) does not depend on model.  The MAJOR ERROR is that (1) is NOT equivalent to (3). Instead, it is equivalent to the following:

 min { E_y~d [  log d (y) ]  -  E_y~d  E_x~d(x|y) [ log m (y|x) ]}                     (3’)

Notice the swap of “E_x” and “log”. By Jensen’s nequality, we have

 log E_x~d(x|y)  m (y|x) ]  > E_x~d(x|y) [ log m (y|x)
 -  E_y~d  log E_x~d(x|y)  [ m (y|x) ]    < -  E_y~d  E_x~d(x|y) [ log m (y|x) ]

So, minimizing (3) amounts to minimizing a lower bound of the correct objective (3’). It does not make sense at all.

---

> ### Author Response · Authors · 2018-11-27
> **Response to Reviewer 2**
>
> Thank you for taking the time to review the paper.  We corrected the statement and refactored the paper to reflect this. We hope you will be able to read the revision and verify that your concerns have been addressed. Please let us know if you have any further questions.

---

> > ### Comment · AnonReviewer2 · 2018-11-30
> > **question not answered**
> >
> > The authors modified the paper, but did not explain how my concern is addressed. Although the theory is changed, the objective function remains the same. I am not convinced that the issue is addressed.
> >
> > In addition, the purpose of the rebuttal process is to provide authors with the opportunity to clarify misunderstandings, NOT to change the main theory. If the main theory is flawed, the paper cannot be accepted in this round.

---

> > > ### Author Response · Authors · 2018-11-30
> > > **You complained about subsection 2.1 and we addressed it. The main theory of the paper was not subsection 2.1 of version 1. The main theory of the paper has not changed.**
> > >
> > > Dear Reviewer 2,
> > >
> > > Thank you for replying to the rebuttal.
> > >
> > > 1- Your concern---your whole review---was only about subsection 2.1 of the first version of the paper titled "A peculiarity of maximum likelihood learning". We would like to point out that this subsection was not the "main theory of the paper" as you suggest. The main theory of the paper was and is still to propose a new objective function that mitigates the "input forgetting" issue of maximum likelihood. We proposed this objective for supervised learning and for unsupervised learning with deep latent variable models. We then ran an empirical study which showed the RLL objective outperforms maximum likelihood in a classification under imbalance study and in a neural topic modeling experiment.
> > >
> > > Subsection 2.1 of the first version was about justifying why we propose the objective as a potential fix for the "input forgetting" issue of maximum likelihood.
> > >
> > > 2- We addressed your concern by providing another justification for why the objective we propose makes sense. We do this in the revision in the paragraph of the introduction titled "the finite sample dilemma of maximum likelihood".
> > >
> > > If there is one thing to get out of that paragraph it is this: Eq. 6 shows that a learning algorithm may find it easier to increase the left hand side of the equality---the maximum likelihood objective---by increasing the first term rather than decreasing the second term. The problem is that the second term is the only term with information regarding how outputs y relate to inputs x. A natural thing to do to avoid this is to use the proposed RLL objective which penalizes maximization of the first term in the right hand side of Eq. 6.
> > >
> > > 3- The reason why we restructured the paper is that the other reviewers complained that our paper as is was confusing. However this restructuring did not change "the main theory of the paper" which is (1) the new objective, (2) what it is for supervised learning, (3) what it is for unsupervised learning, and (4) its comparison to maximum likelihood in an empirical study.
> > >
> > > If you are concerned that it is not clear why the RLL objective works, we explain this in the introduction of the revision. We also point you out to our message above regarding the connection of RLL to pointwise mutual information (PMI).
> > >
> > > Thanks for your reply.

---

> > > > ### Comment · AnonReviewer2 · 2018-11-30
> > > > **your original text**
> > > >
> > > > In the original version, you wrote: "Contributions. We identify a peculiarity in maximum likelihood learning that causes the input forgetting problem in Section 2.1. We then propose a new learning criterion to mitigate this issue."  The intend was to address "a peculiarity in ML learning".  But, we are addressing something else. This is a change of main theory (story) to me.

---

> > > > > ### Comment · AnonReviewer2 · 2018-11-30
> > > > > **typo**
> > > > >
> > > > > But you are now addressing something else.

---

> > > > > > ### Author Response · Authors · 2018-11-30
> > > > > > **We are addressing the same problem.**
> > > > > >
> > > > > > Dear Reviewer 2,
> > > > > >
> > > > > > We are glad that you want to get to the bottom of the problem we are trying to address in the paper.
> > > > > >
> > > > > > We are still addressing a peculiarity in maximum likelihood that causes "input forgetting". This peculiarity was wrongly stated in subsection 2.1 of version 1 of the paper. We are now highlighting this peculiarity in the introduction in the paragraph titled "the finite sample dilemma of maximum likelihood".
> > > > > >
> > > > > > The main story of the paper is to say "look there is this problem that keeps happening when fitting deep models with the maximum likelihood objective. This problem manifests itself in deep latent variable models as posterior collapse. This problem manifests itself in Seq2Seq conversation models as production of generic responses by the decoder. This problem manifests itself as failure to predict rare classes in classification under imbalance. All these issues can be nailed down to one common issue: the variables being conditioned upon are not taken into account by the neural network. In deep latent variable models the conditioning variable is the latent variable. In SeqSeq it is the output of the encoder. In classification, it is the input covariate x. We propose this objective that ties outputs to conditioning variables by subtracting a marginal to the original maximum likelihood objective. We think this objective should fix the problem. How does this objective fix the problem? We suggest looking at the KL divergence formulation of maximum likelihood (the problematic subsection 2.1 of version 1). We define the marginal in the proposed objective in supervised learning. We define the marginal in unsupervised learning as well. We now look at how this objective compares to maximum likelihood in a supervised learning problem and an unsupervised learning problem. We see that the objective outperforms maximum likelihood."
> > > > > >
> > > > > > This is the same story both in version 1 and in the revision. What changed is the answer to the question "How does this objective fix the problem?". The response we provided in subsection 2.1 of version 1 was not correct. We fix this by using the "finite sample dilemma of maximum likelihood" argument in the introduction.
> > > > > >
> > > > > > Further evidence of why the RLL objective works is its relationship to pointwise mutual information, KL divergences, and ranking.

---

### Official Review · AnonReviewer1 · 2018-11-01
**Interesting ideas that need further refinement**

**Rating:** 4
**Confidence:** 4

**Review:**

Summary:

This paper proposes maximizing the “reflective likelihood,” which the authors define as: E_x E_y [log q(y|x) - \alpha log q(y)] where the expectations are taken over the data, q is the classifier, and \alpha is a weight on the log q(y) term.  The paper derives the reflective likelihood for classification models and unsupervised latent variable models.  Choices for \alpha are also discussed, and connections are made to ranking losses.  Results show superior F1 and perplexity in MNIST classification and 20NewsGroups modeling.

Pros:

I like how the paper frames the reflective likelihood as a ranking loss.  It does seem like subtracting off the marginal probability of y from the conditional likelihood should indeed ‘focus’ the model on the dependent relationship y|x.  Can this be further formalized?  I would be very interested in seeing a derivation of this kind.

I like that the authors test under class imbalance and report F1 metrics in the experiments as it does seem the proposed method operates through better calibration.

Cons:

My biggest issue with the paper is that I find much of the discussion lacks rigor.  I followed the argument through to Equation 3, but then I became confused when the discussion turned to ‘dependence paths’: “we want our learning procedure to follow the dependence path—the subspace in Θ for which inputs and outputs are dependent. However this dependence path is unknown to us; there is nothing in Eq. 1 that guides learning to follow this dependence path instead of following Eq. 3—the independence path” (p 3).  What are these dependence paths?  Can they be defined mathematically in a way that is more direct than switching around the KLD directions in Equations 1-3?  Surely any conditional model x-->y has a ‘dependence path’ flowing from y to x, so it seems the paper is trying to make some stronger statement about the conditional structure?

Moving on to the proposed reflective likelihood in Equation 4, I could see some connections to Equations 1-3, but I’m not sure how exactly that final form was settled upon.  There seems to be a connection to maximum entropy methods?  That is,   E_x E_y [log q(y|x) - \alpha log q(y)] = E_x E_y [log q(y|x)] + \alpha E_y [ -log q(y)] \approx E_x E_y [log q(y|x)] + \alpha H[y], if we assume q(y) approximates the empirical distribution of y well.  Thus, the objective can be thought of as maximizing the traditional log model probability plus an estimate of the entropy.  As there is a long history of maximum entropy methods / classifiers, I’m surprised there were no mentions or references to this literature.  Also, I believe there might be some connections to Bayesian loss calibration / risk by viewing \alpha as a utility function (which is easy to do when it is defined to be data dependent).  I’m less sure about this connection though; see Cobb et al. (2018) (https://arxiv.org/abs/1805.03901) and its citations for references.

The data sets used in the experiments are also somewhat dissatisfying as MNIST and 20NewsGroups are fairly easy to get high-performing models for.  I would have liked to have seen more direct analysis / simulation of what we expect from the reflective likelihood.  As I mentioned above, I suspect its really providing gains through better calibration---which the authors may recognize as F1 scores are reported and class imbalance tested---but the word ‘calibration’ is never mentioned.  More direction comparison against calibration methods such as Platt scaling would be make the experiments have better focus.  It would be great to show that this method provides good calibration directly during optimization and doesn’t need the post-hoc calibration steps that most methods require.

Evaluation:  While the paper has some interesting ideas, they are not well defined, making the paper unready for publication.  Discussion of the connections to calibration and maximum entropy seems like a large piece missing from the paper’s argument.

---

> ### Author Response · Authors · 2018-11-27
> **Response to Reviewer 1**
>
> Thank you for your in depth review. We are very happy that you enjoyed the connections we made to ranking losses. We think this was a cool finding as well! In fact there might be more connections: the coefficient \alpha can be data-dependent and in that sense it induces a family of regularizers. The step-schedule for alpha (as defined in Eq. 11 of the paper) corresponds to ranking loss. We conjecture that there might be other connections when carefully choosing other schedules for \alpha.
>
> We think the “finite sample dilemma” derivations address your remark about formalizing the fact that subtracting the marginal will make the optimization focus on capturing the dependencies. Please let us know if you think otherwise.
>
> We agree with you on the calibration remark. We also found classification under imbalance to be a natural fit for testing the RLL objective. The results suggest RLL is doing what it is supposed to be doing i.e. strengthen the dependence between x and y.
>
> Regarding your remarks on terminology and lack of rigor in the discussion: we clarified the exposition of the paper and formalized the explanation as to why the proposed objective works. Please see the general rebuttal above for details on what changed in this new version. We also hope you will find the time to read the revision.
>
> Regarding your comment on connections: thank you for bringing this up! Although it might seem that there is a connection to maximum entropy methods, there is no such connection unfortunately. We would like to point out that in your derivation, the quantity you define as entropy is not the entropy of q(y). This is because the expectation is not taken under q(y) but under p*(y | x)---the conditional distribution of y given x under the population distribution---which is different from q(y). This is why we did not mention connections to maximum entropy methods. Thank you for the reference! We have not looked into connections to Bayesian loss calibration.
>
> We hope we have addressed your concerns. Please let us know if you have other remarks. We would appreciate it if you could read the revision and let us know if we have addressed all your concerns.

---

> > ### Comment · AnonReviewer1 · 2018-12-01
> > **Re: Response to Reviewer 1**
> >
> > Thanks for your responses and revisions, authors.  I do find the finite sample derivation in Equations 3-6 of the new draft to be clearer.  It is an interesting observation.
> >
> > Unfortunately, my "biggest issue" of the lack of rigor still stands.  While the derivations are rigorous in the new draft (i.e. I follow the algebra), I don't find the motivations and logic behind them to be.  I don't find them to have a clear flaw per se, but I find them hand-wavy.  Overall, it's very hard for me to swallow that there is a deficiency in maximum likelihood learning and that Equation 7 fixes this deficiency in just ~two pages of exposition.  Moreover, there are still no theorems, clear mathematical definitions, or some simulations.  For instance, can you show how the reflective likelihood changes analytically tractable / closed-form solutions?  What would the 'reflective OLS estimator' be?  These simpler, classical cases need addressed before I could be convinced that it fixes the problem.

---

> > > ### Author Response · Authors · 2018-12-03
> > > **Thank you for your insightful feedback.**
> > >
> > > Thank you Reviewer 1 for your insightful comments. We answer your questions below.
> > >
> > > 1--"I don't find the motivations and logic behind the derivations to be rigorous".
> > >
> > > We explain the motivation behind the derivations as follows. Consider supervised learning with input/output pairs (x, y). Input forgetting happens--by definition--when the input x is not being taken into account by the neural network to predict the output y. This led us to consider looking at the maximum likelihood objective for a conditional model p_{\theta}(y | x) but with an independent assumption on x and y. This leads to Eq. 5 in the draft. However fitting Eq. 5 corresponds to fitting a marginal distribution over y. This is input forgetting. We then relate the objective in Eq. 5 to the maximum likelihood objective of interest. This leads to Eq. 6 which shows that performing maximum likelihood corresponds to a tradeoff between fitting the marginal (the first term in Eq. 6) or minimizing the second term (this second term is the only term that contains information that relates x and y). However a learning algorithm might find it easier to fit the first term (the marginal over y) than to minimize the second term. We make sure to prevent fitting the marginal by regularizing maximum likelihood .
> > >
> > > 2--"Overall, it's very hard for me to swallow that there is a deficiency in maximum likelihood learning and that Equation 7 fixes this deficiency in just two pages of exposition."
> > >
> > > We are proposing to regularize maximum likelihood learning to impose a stringer dependence between variables. We are not throwing away maximizing log likelihood completely. The RLL objective is the difference between the log likelihood and the log reflective probability of the outputs y. Our work is about noticing a common problem when fitting deep models with log likelihood and proposing a potential solution to fix the problem. There have been many methods proposed to regularize maximum likelihood. These "penalized maximum likelihood" objectives include Lasso (which regularizes maximum likelihood by penalizing the L1 norm of the parameters) and Ridge also known as weight decay in the deep learning literature (which regularizes maximum likelihood by penalizing the L2 norm of the parameters). The method we propose is a data-dependent regularization method that regularizes maximum likelihood by minimizing the log reflective probability of the outputs. There are even more alternatives to maximum likelihood in the statistics literature. See for example Generalized Estimating Equations (GEE) for longitudinal data.
> > >
> > > We now provide further evidence that the proposed RLL objective promotes a stronger dependence between inputs and outputs. Consider a fixed \alpha schedule \alpha_n = \alpha_0 for all n and 0 < \alpha_0 < 1, we can show the RLL objective in Eq. 8 can be rewritten as:
> > >
> > > L_{RLL} = 1/N \sum_{1}^{N} [ (1 - \alpha_0) * \log p_{\theta}(y_n | x_n) + \alpha_0 * PMI(x_n, y_n) ]
> > >
> > > where PMI(x_n, y_n) = \log p_{\theta}(y_n | x_n) - \log p_{\theta}^{refl}(y_n)
> > >
> > > Essentially this shows that RLL regularizes maximum likelihood by maximizing the pointwise mutual information between individual pairs (x_n, y_n). This is the reason why it promotes a stronger dependence between inputs and outputs.
> > >
> > > 3--"Moreover, there are still no theorems, clear mathematical definitions, or some simulations."
> > >
> > > We did not think a theorem was needed. For example the consistency of the objective in Eq. 8 is a simple consequence of applying the law of large numbers and invoking the continuity of logarithm.
> > >
> > > However we would like to hear about what type of theorem you were expecting.
> > >
> > > 4--"For instance, can you show how the reflective likelihood changes analytically tractable / closed-form solutions?  What would the 'reflective OLS estimator' be?  These simpler, classical cases need addressed before I could be convinced that it fixes the problem."
> > >
> > > We looked into this as well. In fact there is no closed form formula for the RLL objective on the usual linear model. One has to solve an estimating equation to find the RLL solution. This is expected because RLL uses a data-dependent regularizer.
> > >
> > > Please let us know if our response answers your comments above. We are looking forward to your reply.

---

### Comment · AnonReviewer3 · 2018-10-25
**please clarify**

You write "Maximizing Eq. 1 can be achieved by maximizing either Eq. 2 or Eq. 3 or both", and this seems to be crucial to your motivation for the proposed method. However this statement is false. It's true that the true conditional model p(y|x) is a solution to eq 1,2 and 3, but the converse does not hold: There are many solutions to equation 3 that do not maximize equation 1. You are basically claiming that maximum likelihood is not a consistent estimation method, contradicting all of the statistical literature. Please clarify your motivation for the proposed method, and let me know if I'm misunderstanding.

---

> ### Author Response · Authors · 2018-10-25
> **Clarifications**
>
> Thank you AnonReviewer3 for your feedback and for giving us the opportunity to clarify things before you make your decision! There are several points raised in your comment that we would like to provide an answer to.
>
> 1. "... this statement is false. It's true that the true conditional model p(y|x) is a solution to eq 1,2 and 3, but the converse does not hold..."
>
> We agree with you that  statement is misleading and wrong as formulated. It was not meant as a statement on the maxima (clearly Eq. 1 and Eq. 3 may not have the same global maxima; there are simple counterexamples to show this), but as a way to distinguish which paths are followed to maximize Eq. 1. Let us clarify what we mean by this as follows:
>
> Consider a supervised learning setting. We have observations (x, y) and our goal is to fit a conditional model p_{\theta}(y | x). We can fit this model by maximizing either (1) or (2) as they are equivalent; they correspond to maximum likelihood estimation. However, in practice, when we learn a conditional model p_{\theta}(y | x) that is arbitrarily flexible (e.g., parameterized by a deep neural network) by optimizing (1) (or equivalently (2)) using data, the resulting model often has the issue that it ignores the inputs x. In this case, the resulting value of the parameters is analogous as if we had optimized (3) instead. In other words, when optimizing (1) with data, nothing guarantees that (3) is not being optimized. This behavior is undesirable. To prevent that, we want to promote a strong dependency between y and x. That is, we propose to avoid the "marginal path", as induced by (3).
>
> We will edit out the statement in the revision and make that part of the paper more clear.
>
> 2. "You are basically claiming that maximum likelihood is not a consistent estimation method, contradicting all of the statistical literature."
>
> If by "consistent estimation method" you mean consistency in the statistical sense (i.e. convergence in probability of the estimator to the true parameter as sample size goes to infinity) then no we are not studying consistency/inconsistency of maximum likelihood in the paper.
>
> However we would like to point out that maximum likelihood does not always lead to consistent estimators. Consider the counterexample of Bahadur, 1958 (see [1] for the reference) showing an example where maximum likelihood is inconsistent.
>
> 3. "Please clarify your motivation for the proposed method, and let me know if I'm misunderstanding."
>
> Thank you for giving us the opportunity to make things more clear. Our motivation for the paper is this: there is this common behavior we call “input forgetting” in the paper that happens quite often with models parameterized by deep neural networks. This is manifested in deep latent variable models as the phenomenon known as “posterior collapse” or “latent variable collapse” in the literature. This also happens in RBMs (see [2]). However the problem can happen even without latent variables. Some other examples we have not mentioned in the paper include Seq2Seq models where the decoder does not account for the input. A good manifestation of this is in neural conversation models where the decoder provides very generic responses such as “I don’t know” or “ok” no matter what the query/input is. See for example [3] for more details on generic answers in conversation models.
>
> One common denominator of all these examples is that the variable being conditioned upon is ignored by the deep network. Our paper proposes a regularization approach to mitigate this problem. We add a regularizer termed the “reflective likelihood” that is basically a marginal distribution over the output variable. We define this marginal in the paper for both supervised and unsupervised learning. The resulting objective is the difference between the usual maximum likelihood objective and this reflective likelihood. Subtracting the reflective likelihood forces the optimization to favor parameter settings that promote usage of the variable being conditioned upon. We validate this hypothesis through our empirical studies where we notice an improvement in terms of latent variable collapse and classification performance for rare classes.
>
> In summary: for applications where you care about promoting a stronger dependence between inputs and outputs (e.g. in deep latent variable models or in classification under imbalance) then we propose to use the objective proposed in this paper instead of vanilla MLE.
>
> We hope our answer clarifies things. Thank you for bringing these points up. We will add these clarifications in the revision.
>
> [1] R. R. Bahadur. Examples of Inconsistency of Maximum Likelihood Estimates. The Indian Journal of Statistics, 1958.
> [2] K. Cho et al. Enhanced Gradient and Adaptive Learning Rate for Training Restricted Boltzmann Machines. In ICML,2011.
> [3] J. Li et al. A Diversity-Promoting Objective Function for Neural Conversation Models. In NAACL, 2016.

---

### Author Response · Authors · 2018-11-27
**Rebuttal: In response to reviewer feedback we refactored the paper. The idea is the same, the results are the same, the exposition is different.**

We thank all the reviewers for taking time to review our paper. Your feedback has greatly helped us revise the paper. The two main concerns from all three reviewers were that (1) the equivalence statement made in section 2 of the first version of the paper was incorrect and (2) the paper is very confusing.  We rewrote the paper to address these two issues. We corrected the equivalence statement and refactored the paper to further clarify the intuitions and technical details behind our proposed idea.  We hope the reviewers will read the revision. We apologize for taking time to post the revision. We wanted to make sure we addressed all the concerns.

We want to draw attention on the importance of the issue tackled by the paper. The most ubiquitous learning objective for deep models is maximum likelihood. It works very well in practice in most cases. However there are cases where maximum likelihood leads to poor behavior. For example it leads to posterior collapse in deep latent-variable models. Furthermore, it causes lack of diversity in generated responses in Seq2Seq conversation models. Finally, it struggles to learn useful features for rare classes in classification when the class distribution is highly skewed. All these issues can be summarized into one main behavior: the variables being conditioned upon are not taken into consideration by the deep network. We call this problem "input forgetting". We identify a potential cause of this issue and propose the RLL objective to alleviate it.

The new structure of the paper is as follows:

1- We state the "finite sample dilemma" of maximum likelihood in the introduction. This replaces the section 2 of the first version of the paper.  We mention our contributions and related work also in the introduction.

2- In section 2 of this current version we derive the RLL objective for supervised learning and propose a practical stochastic approximation of it.

3- In section 3 we extend RLL to unsupervised learning using the auto-encoding framework...this leads us to proposing reflective auto-encoders (RAEs)---a new family of stochastic auto-encoders that do not suffer from posterior collapse.

4- We discuss connections to ranking losses in a new section 4.

5- We finally present empirical findings in section 5. To save space, we added the table listing the learned topics to the appendix.

---

### Author Response · Authors · 2018-11-28
**New insights on the RLL objective: relationship to pointwise mutual information and KL divergences**

We extended section 4 with new findings. We unfortunately cannot post a new revision at the moment. We will add the new revision once it is allowed to upload revisions again.

We found the RLL objective in Eq. 8 can be rewritten as a convex combination between the log likelihood \log p_{\theta}(y_n | x_n) and the pointwise mutual information between x_n and y_n for each data pair (x_n, y_n) when the schedule for alpha is \alpha_n = \alpha_0 < 1. This new perspective is yet another proof that RLL promotes a stronger dependence between inputs and outputs. The pointwise mutual information term forces each output y_n to strongly depend on its corresponding input x_n. This justifies the huge gain in performance in classification under imbalance. Note pointwise mutual information is stronger than mutual information when it comes to dependence. This is because pointwise mutual information acts at the datapoint level whereas mutual information is an average.

We also found that for a fixed \alpha schedule the RLL objective can be written as a difference of KL divergences: \alpha_0 KL(p_{data}(y_n | x_n) || p_{\theta}^{refl}(y_n)) - KL(p_{data}(y_n | x_n) || p_{\theta}(y_n | x_n)). Maximizing the RLL is equivalent to  fitting the conditional model p_{\theta}(y_n | x_n) on the data while "unfitting" the unconditional model defined by the reflective probability  p_{\theta}^{refl}(y_n) on the data.

---

### Meta-Review · Area_Chair1 · 2018-12-14
**Interesting problem and approach, but not quite rigorous enough.**

**Confidence:** 4
**Recommendation:** Reject

**Metareview:**

The proposed “input forgetting” problem is interesting, and the reflective likelihood can come to be seen as a natural solution, however the reviewers overall are concerned about the rigor of the paper. Reviewer 2 pointed out a technical flaw and this was addressed, however the reviewers remain unconvinced about the theoretical justification for the approach. One suggestion made by reviewer 1 is to focus on simpler models that can be studied more rigorously. Alternatively, it could be useful to focus on stronger empirical results. The method works in the experiments given, but for example in the imbalanced data experiments, only MLE is compared to as a baseline. I think it would be more convincing to compare against stronger baselines from the literature. If they are orthogonal to the choice of estimator, then it would be even better to show that these baselines + RLL outperforms the baselines + MLE. Alternatively, you mention some challenging tasks like seq2seq, where a convincing demonstration would greatly strengthen the paper. While the paper is not yet ready in its current form, it seems like a promising approach that is worth further exploration.